# Geographical Origin Has a Greater Impact on Grape Berry Fungal Community than Grape Variety and Maturation State

**DOI:** 10.3390/microorganisms7120669

**Published:** 2019-12-10

**Authors:** Dimitrios Kioroglou, Elena Kraeva-Deloire, Leigh M. Schmidtke, Albert Mas, Maria C. Portillo

**Affiliations:** 1Depertment Bioquímica i Biotecnologia, Facultat d‘Enologia, Rovira i Virgili University, 43007 Tarragona, Spainalbert.mas@urv.cat (A.M.); 2National Wine and Grape Industry Centre, Charles Sturt University, Wagga Wagga, NSW 2678, Australia; lavandinia@gmail.com (E.K.-D.); lschmidtke@csu.edu.au (L.M.S.); 3School of Agricultural and Wine Sciences, Charles Sturt University, Wagga Wagga, NSW 2650, Australia

**Keywords:** grape microbiome, ripening state, massive sequencing, biogeography, GNEISS balance

## Abstract

We used barcoded sequencing to analyze the eukaryotic population in the grape berries at different ripening states in four Australian vineyards. Furthermore, we used an innovative compositional data analysis for assessing the diversity of microbiome communities. The novelty was the introduction of log-ratio balances between the detected genera. Altogether, our results suggest that fungal communities were more impacted by the geographical origin of the Australian vineyards than grape variety and harvest time. Even if the most abundant genera were *Aureobasidium* and *Mycosphaerella*, they were ubiquitous to all samples and were not discriminative. In fact, the balances and the fungal community structure seemed to be greatly affected by changes of the genera *Penicillium*, *Colletotrichum*, *Aspergillus*, *Rhodotorula*, and *Botrytis*. These results were not evident from the comparison of relative abundance based on OTU counts alone, remarking the importance of the balance analysis for microbiome studies.

## 1. Introduction

The concept of “terroir,” in oenology refers to a geographic area characterized mainly by its climate, soil, and human factors that contribute to producing typical wines. However, the term “microbial terroir” is recently gaining interest in viticultural studies to indicate the importance of the vineyard microbiome composition over the regional wine typicity. The grapevine microbiome is the complex community of microorganisms, including fungi and bacteria that interact with the whole plant and play a key role in plant health, growth, and nutrient uptake [1].

Recently, due to advances in metagenomics and the development of high-throughput sequencing (HTS) techniques, the grapevine microbiome is receiving increasing attention. Metagenomic analyses suggest that the microbial communities associated with grapes and grape must resemble the ones present on leaves [2,3] and have their source mainly in the soil and surrounding fields [4]. Furthermore, grape microorganisms can be transferred to the winery where, ultimately, they may affect wine chemical composition and influence its quality, even at the regional scale [5,6,7,8].

Several factors and vineyard characteristics have recently shown that grape microbiome is influenced by vineyard characteristics like climate, region, site, and grape cultivar, suggesting that there is a nonrandom microbial fingerprint associated with the terroir [2,3,9,10,11,12,13,14]. One important factor that has been proven to induce changes in grape microbiome composition and structure is the berry development process [15,16]. The grape ripening stages are defined mainly by physiochemical changes, such as increase in levels of phenolic compounds and accumulation of sugars [17]. Large numbers of yeast species have been identified on grape berries with population densities ranging from 101 to 103 CFU/g (Colony Forming Units) on immature grapes but increasing to 103–106 CFU/g at harvest time [9,15,18,19,20,21]. However, previous studies analyzing microbiome changes through berry maturation have been mostly based on culture-dependent techniques that have been proven to be insufficient to reveal the environmental microbial diversity and ecology [22,23]. As a consequence, currently little is known about the real influence of the grape berry maturation state on its microbiome.

One of the major drawbacks of the studies analyzing microbiome diversity by means of HTS techniques is the derivation of statistical inferences after converting the OTU (operational taxonomic unit) counts of the identified genera to relative abundance. The transformation of the OTU counts to compositional data, on one hand, adds the constraint of the abundances having to sum to a constant (i.e., 1), and on the other hand, may lead to misinterpretations when multivariate statistics are applied [24]. Moreover, the nature of compositional data is known to hinder proper differential abundance analysis since various normalization methods and statistical assumptions could potentially not be appropriate for this type of data [25]. Therefore, in the current study inferences on the differences of the microbial communities have been derived by using the compositional analysis toolbox GNEISS [26], as incorporated into Quantitative Insights Into Microbial Ecology framework (QIIME version 2019.1) [27]. GNEIIS introduces the concept of balances which refer to the log-ratio between specific microbial subsets of the community, eliminating the need of using relative abundances and statistical assumptions.

In this work, we use HTS to investigate the fungal biota (mycobiome) composition at two maturation stages of grape berries from two grapevine cultivars, Cabernet Sauvignon and Syrah, growing under two different geographical and environmental conditions. We also compare the usefulness of balances obtained by the GNEISS toolbox for the microbial diversity analysis on our data set.

## 2. Materials and Methods 

### 2.1. Experimental Vineyard and Harvesting

Grapes were sourced from two Australian wine regions sampled during 2015 and denominated Griffith (G) (Riverina, New South Wales, Australia) and Orange (O) (Coordinates and elevation in Appendix A). These wine regions represent two distinctively different grape growing regions. The Griffith region is classified as a warm to very warm grape growing area with temperate nights [28] and is characterized by a flat terrain (around 130 m above sea level, a.s.l.) and secure water supply, enabling it to maintain a 15% share in the total Australian grape production. In contrast, the Orange (O) region has an undulating to mountainous terrain with vineyard elevations spanning from 600 up to 1000 m a.s.l. The Orange region is classified as temperate to temperate/warm with cool to very cool nights. Two commercial vineyards were selected in both regions (designated G1, G2, O1, and O2), for Shiraz vines (S) whereas Cabernet Sauvignon (C) vines were sampled only at vineyard G1 and O2. 

Between G1 and G2 there is less than 5 m altitude difference whereas the O1 site is at 607 m a.s.l. and the O2 site at 876 m a.s.l. thus having an approximate difference of 270 m (Appendix A). Both S and C vines were own rooted, grown under drip irrigation, and trellised to a sprawling system in G. In O, vines were trellised to vertical shoot positioning. The nitrogen management throughout the season was similar for both cultivars and the average crop yields of both plots were approximately 15−20 tons per hectare. During the season, mesoclimatic temperatures, stem water potential, and soil moisture were monitored in an attempt to characterize experimental plots. Harvest dates for vineyards (H1 and H3) were determined at the point where sugar accumulation per berry and berry fresh mass in conjunction with °Brix. The first harvest for both cultivars occurred at approximately 21 °Brix and was designated H1. The second harvest, designated as H3, occurred at 23 °Brix for both cultivars. At each harvest date, 60 kg of grapes was randomly harvested across the vineyards for each variety with an addition of 40 mg/kg of potassium metabisulfite prior to transport to the Charles Sturt University (CSU)/ National Wine and Grape Industry Centre (NWGIC) experimental winery. On arrival, a 100-berry subsample from each replicate was collected and immediately frozen at −20 °C for further analyses with a total of 50 subsamples.

### 2.2. DNA Extraction

Must samples were defrosted and centrifuged at 3500 g for 15 min, washed three times with ice cold phosphate buffered saline and the pellet resuspended in 200 µL of DNeasy lysis buffer (Qiagen, Valencia, CA, USA) supplemented with 40 mg/mL lysozyme and incubated at 37 °C for 30 min. After this point, DNA extraction continued following the protocol of the QIAmp Fast DNA Stool Mini Kit (Qiagen,), with the addition of a bead beater cell lysis step for 2 min using a FastPrep-24 (MP Bio) and 100 µL of DNA eluted using AE buffer (Qiagen). DNA concentration and quality were assessed using a Quantus Fluorometer (Promega, Madison, WA, USA) followed by gel electrophoresis of 5 µL of eluant in 1.5% agarose submerged in 1X TAE buffer. Gels were stained with GelRed™ (Biotium, Fremont, CA, USA) nucleic acid gel stain and viewed under UV light using Gel Doc XR+ Imaging system (Bio-Rad, Hercules, CA, USA) DNA samples (approx. 70 ng) were subject to PCR amplification and sequencing performed by the Australian Genome Research Facility. PCR amplicons were generated using as forward primer (ITS1: CTTGGTCATTTAGAGGAAGTAA or 341 F: CCTAYGGGRBGCASCAG) and reverse primer (ITS2: GCTGCGTTCTTCATCGATGC or 806 R: GGACTACNNGGGTATCTAAT), with 35 cycles, an initiation temperature of 95 °C for 7 min, disassociate conditions of 94 °C for 30 sec, annealing conditions of 55 °C for 45 sec for ITS or 50 °C for 60 sec for 16S RNA, extension at 72 °C for 60 sec and a final temperature of 72 °C for 7 min. Thermocycling was completed with an Applied Biosystem 384 Veriti and using AmpliTaq Gold 360 mastermix (Applied Biosystems, Foster City, CA, USA) for the primary PCR. The first stage PCR was cleaned using magnetic beads, and samples were visualized on 2% Sybr Egel (Thermo-Fisher, Carlsbad, CA, USA). A secondary PCR to index the amplicons was performed with TaKaRa Taq DNA Polymerase (Takara Shuzo, Otsu, Japan). The resulting amplicons were cleaned again using magnetic beads, quantified by fluorometry by the Promega Quantifluor ST fluorometer (Promega), and normalized. The eqimolar pool was cleaned a final time using magnetic beads to concentrate the pool and then measured using a High-Sensitivity D1000 Tape on an Agilent 2200 TapeStation. The pool was diluted to 5nM and molarity was confirmed again using a High-Sensitivity D1000 Tape. This was followed by sequencing on a MiSeq platform (Illumina, San Diego, CA, USA) with a V3, 600 cycle kit (2 × 300 base pairs paired-end).

Paired-ends reads were assembled by aligning the forward and reverse reads using PEAR (version 0.9.5) [29]. Primers were identified and trimmed. Trimmed sequences were processed using Quantitative Insights into Microbial Ecology (QIIME 1.8) [30] USEARCH [31] (version 8.0.1623) and UPARSE software (version 8.1.1861) [32].

Using USEARCH tools sequences were quality filtered, full length duplicate sequences were removed and sorted by abundance. Singletons or unique reads in the data set were discarded. 

The16S rRNA sequences were clustered followed by chimera filtering using “rdp_gold” database as the reference. To obtain the number of reads in each OTU, reads were clustered with a minimum identity of 97%. Using Qiime taxonomy was assigned using Greengenes database (version 13_8, Aug 2013).

ITS sequences were clustered followed by chimera filtering using “Unite” database as reference. To obtain number of reads in each OTU, reads were clustered with a minimum identity of 97%. Using Qiime, taxonomy was assigned based on Unite database [33] (Unite Version7.1 Dated: 22.08.2016).

### 2.3. Data Analysis

The data processing and part of the statistical analysis has been performed with QIIME (version 2019.1). The OTU table has undergone a series of filtering steps including removing OTUs with < 10 counts across all samples, removing OTUs whose assigned taxonomy did not reach genus level and removing genera whose relative abundance was < 1% across all samples. After collapsing the OTU table at genus level, in order to compensate for the uneven sequencing depth across the samples, the OTU table was rarefied at a value equal to the maximum amount of sequences observed across all samples so as each sample to include 11,579 sequences.

### 2.4. Statistical Analysis

The factors considered for the statistical analysis were region (G1, G2, O1, and O2), variety (C and S) and harvest period (H1 and H3). Statistical analysis has been performed in QIIME with the ADONIS permutation-based statistical test [34] and GNEISS, as well as externally using the Python (version 3.7) libraries STATSMODELS [35], SCIPY [36], and PANDAS [37]. Using the rarefied OTU table, alpha diversity was calculated based on the Shannon index, whereas beta diversity was based on the Bray-Curtis index since taxonomy was constrained at genus level. Using the Shannon index, the replicates were examined for outliers resulting in the removal of two samples. The distribution of the Shannon index proved of being bimodal, with one mode concerning only the region O1. The two modes were separated and two-way ANOVA was performed on each mode. Prior to ANOVA, the assumptions of heteroskedasticity and normality on each mode were examined and satisfied using the Levene and Shapiro-Wilk test, respectively. Similarities between regions, varieties, and harvest periods were examined with Principal Coordinate Analysis (PCoA) using the Bray-Curtis distance metric, whereas ADONIS multivariate analysis of variance (MANOVA) with 999 permutations helped to identify significance. The unrarefied OTU table became the input source for GNEISS since it applies its own normalization method. The analytical pipeline for GNEISS included the initial steps of imputing zero OTU counts with a pseudocount equal to 1 and partitioning of genera into two groups using Ward hierarchical clustering. Each group contains genera that are highly correlated based on their co-occurrence and therefore the two groups are anti-correlated. Subsequently, GNEISS applied isometric log-ratio transformation which calculates, for each sample, the log-ratio between these two groups. That means one group represents the numerator of the ratio and the other group the denominator, whereas the log-ratio is referred as balance. This balance may have positive or negative value signifying that for a given sample the abundances of some genera have changed and these genera are either from the numerator, the denominator, or both the numerator and denominator. Finally, based on these balances GNEISS performed ordinary least squares regression (OLS) using the regressors region, variety, and harvest period, where 10-fold cross validation of 10 partitions showed that over fitting did not occur. Since OLS regression is more appropriate for continuous than categorical independent variables, from the reported results the only statistical measure considered was the explained variance (*R*^2^ adjusted).

## 3. Results

### 3.1. Sequences Analysis Results

DNA of 50 samples of grapes representative of the 4 vineyards (G1, G2, O1, and O2) were massively sequenced by the Illumina platform resulting in a minimum of 30K reads for both the ITS and 16S rRNA gene regions, respectively. After quality filtering and exclusion of sequences matching to chloroplast or mitochondria, 1,187,046 and 24,610 reads remained for fungal and bacterial community analysis, respectively. In the case of ITS sequences the median number of sequences per sample was 23,028 whereas for the 16S rRNA sequences it was 366. Given the low number of 16S rRNA sequences per sample combined with the fact that the majority of these sequences have been identified of belonging to the genus *Sphingomonas*, we considered that the obtained sequences for this amplicon were not enough for a robust analysis of the bacterial community.

### 3.2. Fungal Diversity Was Mainly Impacted by the Wine Region

Figure 1 shows alpha diversity of the fungal community based on Shannon index. The Shiraz samples from the Griffith region (G1 and G2) exhibited higher diversity compared to the rest of the samples, whereas the samples with the lowest diversity have been the ones from the O1 region. Additionally, the harvest period (H1 and H3) seemed to have affected the observed diversity of the Shiraz samples from O1. Two-way ANOVA on the Shannon index for the region O1 revealed significant differences between the groups of the factors variety and harvest which had relatively equal amount of impact on the total variance explained by these two factors (54% in total) (Table 1). For the rest of the samples significance was observed only between the groups of the factors region and variety which combined explained 73% of the total variance with the factor region having the greatest impact (53%) (Table 2).

### 3.3. Fungal Community Clustered Distinctly According to Wine Region, Variety and Harvest

The clustering of the samples in the PCoA based on Bray-Curtis distance metric (Figure 2) suggested that the factor region has the greatest effect on the distinction of the samples with O2 samples being the most different. Moreover, higher order of influence was observed on the G2 samples by the factor variety, whereas on the O1 samples the factor harvest was the most influent, as suggested also by the Shannon index. After performing MANOVA with ADONIS on the Bray–Curtis distance metric, the results showed that there are significant differences between the groups of the factors region, variety and harvest period (Table 3). From the three factors, region accounts for the highest amount of variance (53%) followed by variety (7%) and harvest period (2%). 

### 3.4. Genera Balances Affected the Fungal Community Structure 

The high-throughput sequencing analysis allowed the detection of 18 different genera represented by more than 1% of the sequences for each sample. Table 4 shows all the genera that have been identified as well as the range of their OTU counts based on the rarefied OTU table. The most abundant genera in general were *Aureobasidium* and *Mycosphaerella* with a range of OTUs representing 5099 and 1991 on average, respectively. Other abundant genera were *Botrytis, Aspergillus, Colletotrichum, Rhodotorulla,* and *Penicillium*.

Using the tool GNEISS, 9 genera were included by GNEISS in the numerator and 9 in the denominator in order for the balances to be calculated (Figure 3). Furthermore, the range of the OTU counts for each genus is being depicted along with the fold-change between the minimum and maximum observed OTU count. Additionally, for each genus, the samples were grouped based on the factors (region, variety and harvest) and Kruskal–Wallis H-test was performed in order to identify non-significant genera. After applying Bonferroni correction on the resulted *p*-values and setting the significance threshold at 0.01, the genera *Aureobasidium*, *Phoma*, and *Diplodia* were identified as non-significant. Figure 4 shows the balances calculated by GNEISS for each sample showing again that the factor region seems to have the greatest impact on the separation of the samples.

Since GNEISS calculates log-ratio for each sample, it is difficult to conclude whether the calculated log-ratio has been the outcome of changes in the genera of either the numerator, the denominator or both. However, based on the Figure 3 we may assume that the genera that could greatly influence the resulted log-ratio are genera with quite high fold-change between the minimum and maximum observed OTU count. These genera are *Penicillium*, *Colletotrichum*, and *Aspergillus* for the numerator, and *Rhodotorula* and *Botrytis* for the denominator. Figure 5 shows the abundances of the identified genera based on their observed OTU counts. The OTU counts have been converted to relative abundance in order to be compared to their log2 values.

The log2 transformation of the OTU counts in Figure 5 will be used as a means of explaining the calculated balances depicted in Figure 4, taking into consideration the aforementioned genera of Figure 3 that could greatly influence the resulted balance. Starting with the Shiraz samples from G1, the calculated balances are close to zero signifying that the abundances of the genera in the numerator are counter-balanced by the abundances of the genera in the denominator (Figure 4). For instance, the abundances of *Aspergillus*, *Colletotrichum*, and *Penicillium* are relatively equal to those of *Rhodotorula* and *Botrytis*. Therefore, the samples from G1 could conveniently be used as a reference for explaining the change of the balances for the rest of the samples. Consequently, although the abundances of *Aspergillus* and *Penicillium* of the Cabernet samples from G2 are lower than the samples of G1, the higher positive balance of these samples could be attributed to the lower abundance of *Rhodotorula* and *Botrytis* as well as the higher abundance of *Colletotrichum* resulting in an overall more positive balance for G2 samples (Figure 4). Within the G2 Shiraz samples, *Aspergillus* and *Penicillium* had almost the same abundance as the G1 samples, whereas great difference was observed with *Rhodotorula* and *Colletotrichum* (Figure 5). Moreover, the H3 harvest period seems to have influenced positively the abundance of *Aspergillus* and *Colletotrichum*, comparing to H1, and quite negatively the abundance of *Rhodotorula* making these samples the ones with the highest positive balance. All the genera considered for the numerator in the balances had very low relative abundance and log2 of the OTU counts for O1 and O2 samples (Figure 5), resulting in negative balances (Figure 4). As far as the O1 region is concerned, the negative balance was the result of the very low abundances of *Aspergillus*, *Colletotrichum*, and *Penicillium* comparing to G1. Also, comparing the Shiraz and Cabernet samples from O1, the lower abundance of *Aspergillus* and *Colletotrichum* resulted in higher negative balance. Considering the harvest period, the genus that seems to be greatly affected for the Cabernet samples is *Penicillium*, whereas for the Shiraz samples it is *Botrytis*. The Shiraz samples from O2 followed the same pattern as the corresponding samples from O1 (Figure 4), however the higher abundance of *Botrytis* (Figure 5) compared to the rest of the samples was the reason for having the highest negative balance.

The OLS regression performed by GNEISS on the calculated balances using the regressors region, variety and harvest revealed a 75% of total variance explained. From the three factors, region is responsible for the 63% of the total observed variance, whereas variety for 11% and harvest period for 1%.

## 4. Discussion

Microbial communities on grape surfaces have been previously studied due to their perceived importance for contribution to wine characteristics, style, and quality [4,6,14]. Bacterial and fungal populations on the grape surface and the vine plant are affected by various biotic and abiotic factors, such as insects, interactions between resident populations, geography, climate, and viticultural practices [2,4,13]. Generally, many of these variables are not independent and may be clustered into broad groups of effects. Particular attention needs to be paid to the population dynamics of fungi during grape berry development which may be related to the increased surface area of each berry, and to the availability of nutrients such as carbohydrates and organics acids [15,21,38]. Most of the studies analyzing microbial changes during the maturation of the grape have been based on culture-dependent analysis [9,15,18,19,20,21]. Those studies found changes in structure and dynamics of the bacterial and fungal communities during grape maturation. However, the limitation of culture-dependent techniques to assess the real microbial diversity in natural environments is well recognized [22,23]. Recent investigations have characterized significant and consistent changes in grape and wine composition and wine sensory profiles, associated with grape maturities at harvest and vineyard site [39,40]. Besides, higher levels of carotenoids are present in grapes from hot or dry climates, or exposed vineyards to solar radiation [41].

In the present study we used barcoded sequencing to analyze the mycobiome of Cabernet Sauvignon and Shiraz grapes varieties sampled at two ripening times in four vineyards situated at two different Australian wine regions.

Our results show that fungal diversity of the grapes was mainly influenced by region while the varietal and harvest time had a slighter weight. As the climate at O is colder and drier than at G, the pattern of lower fungal species richness in the highest altitude regions hints that selection might have a role in determining these patterns. Within the two O vineyards, greater differentiation between fungal communities was observed than for the G vineyards probably due to a higher heterogeneity of the terrain and differences in altitude (about 270 m) between sampling points O1 and O2. In fact, previous studies have observed changes in microbial diversity and composition due to altitude and geographical orientation [14,42]. In addition, within the O1 samples, the factors variety and harvest had significant influence on the fungal diversity while the weight of harvest or maturation time for the rest of samples was not significant. Except for the absence of *Acremonium* or *Colletotrichium* in most O samples, the list of genera is the same in the rest of samples, thus, the observed changes in alpha diversity should be due to changes in species within each genus. In fact, both genera are usually related to humid or moist climates, which would justify their low abundance in the area O that is qualified as dry.

Our results also show that the fungal community composition varied significantly across the different vineyards. The Bray–Curtis distance metric was used for the clustering of the samples and indicated that the samples clustered significantly different by region, variety and harvest time, being again the region the factor that had the strongest effect on sample differentiation by taxa composition.

The most abundant genera across all samples were *Aureobasidium* and *Mycosphaerella*. While *Aureobasidium* has been frequently isolated and also detected by culture-independent techniques in previous studies analyzing wine grape berries surfaces around the world [2,3,11,16,20,42,43], the genera *Mycosphaerella* has been seldom reported. Because several species of *Mycosphaerella* are considered plant pathogens, the presence of this genus may be directly related to the vine health. However, as different species of the same genus may behave totally different and *Mycosphaerella* was highly abundant in all the analyzed samples, it could represent a characteristic genus of the Australian grapes. In fact, previous studies using HTS analysis to analyze the grape microbiome have also suggested the presence of specific genera or species in different wine regions [2,11,44,45]. Recently, Dissamayake et al. [46] identified both *Aureobasidium* and *Mycosphaerella* within the endophytic community in stems grapevine. Other epiphytic filamentous fungi usually associated with plant diseases and frequently found by HTS in the grape mycobiome were *Aspergillus*, *Botrytis*, *Colletotrichium*, *Rhodotorula,* and *Penicillium*. The results obtained by QIIME corroborated these findings and additionally GNEISS identified these five genera as the ones with the strongest effect on the balances driving the differentiation in fungal composition across samples. Most analyses of microbiome based on HTS relies on multivariate analysis. From the comparison between relative abundance and log2 OTU counts it becomes apparent that the constraint applied when the OTU counts are converted to compositional data could lead to misinterpretations. For instance, the fluctuation of the relative abundance of *Aureobasidium* across all samples seems significant. However, the log2 transformation of the OTU counts of *Aureobasidium* reveals a relatively stable abundance across all samples corroborating its identification as non-significant genus by the current analysis. Thus, the balances analysis applied during the present study lead to more realistic results than those of transforming OTU counts of genera to relative abundance.

## 5. Limitations

Metataxonomic analysis is notorious of incorporating various laboratory and bioinformatic procedures that assign a degree of inaccuracy to the overall analysis by introducing variability among the samples. The numerous factors that are associated with these procedures, along with the fact that the taxonomy and microbial abundance of the samples is unknown render difficult the estimation of the produced variability. Moreover, the nature of the microbial data necessitates the use on non-parametric statistics such as Kruskal-Wallis that do not make assumptions regarding the distribution of the OTU counts. Nevertheless, such non-parametric statistics may lose statistical power in small datasets. Finally, GNEISS does not provide an explicit information regarding differential abundance of taxa. As the results demonstrate, it rather servers as an useful exploratory analysis tool aiming at producing comparative insights between the samples utilizing the concept of balances. This way it sets the basis for a subsequent controlled experimental design that focuses to analysing specific microbial dynamics.

## 6. Conclusions

This study used barcoded massive sequencing to analyze the effect of the grape ripening state, the vineyard region, and grape variety on the grape mycobiome. The results revealed that both fungal composition and diversity were mainly influenced by the vineyard region while the grape variety or the ripening state had less impact. However, within each region the fungal communities were affected differentially by the ripening state apparently due to the climatology. Even if the most abundant genera across samples were *Aureobasidium* and *Mycosphaerella*, the results obtained by GNEISS identified five genera with the strongest effect on the balances driving the changes in fungal composition. This result manifests that the analysis of the microbiome changes based on transformed OTU counts to relative abundance could lead to misinterpretations. 

## Figures and Tables

**Figure 1 microorganisms-07-00669-f001:**
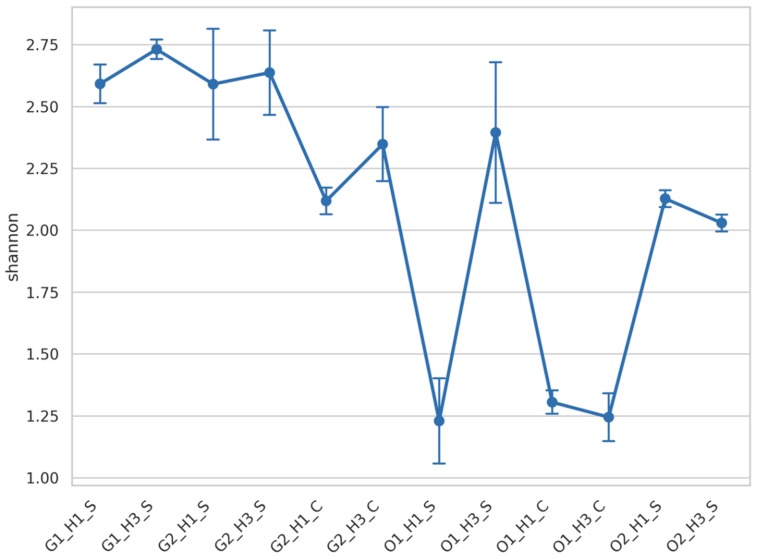
Samples alpha diversity based on Shannon index. Each value corresponds to the average of samples’ replicates and bars correspond to standard deviation. Samples abbreviation includes information of the region, Griffith (G1 and G2) or Orange (O1 and O2), the harvest time points (H1 or H3), and the grape varietals, Shiraz (S) or Cabernet (C).

**Figure 2 microorganisms-07-00669-f002:**
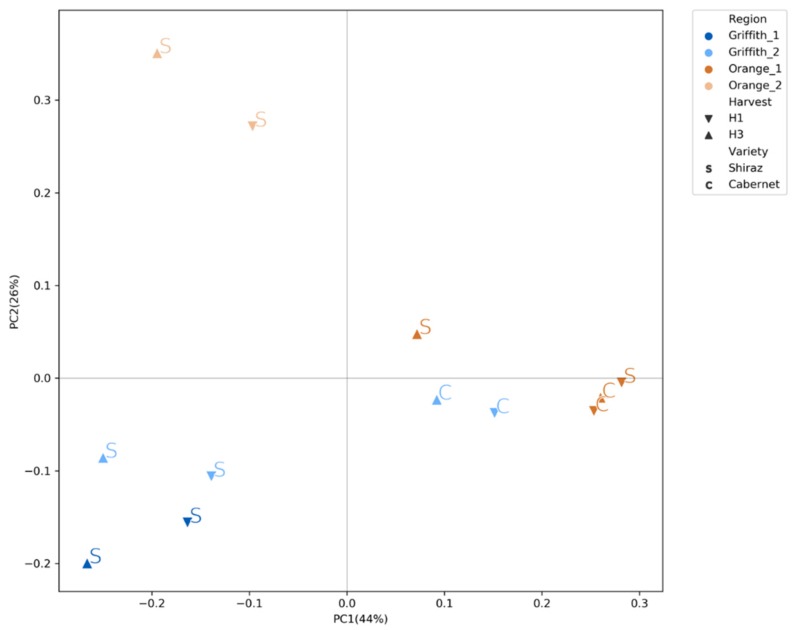
PCoA based on Bray-Curtis distance.

**Figure 3 microorganisms-07-00669-f003:**
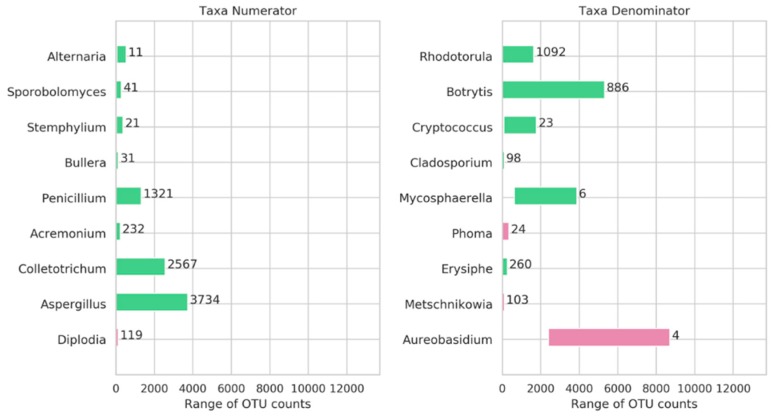
Range of collapsed OTU counts for genera in the numerator and denominator of the balances. OTU counts concern the rarefied OTU table. To the right of each bar, the fold-change between the minimum and maximum observed OTU count is shown. Purple color represents non-significant genera whereas green color significant.

**Figure 4 microorganisms-07-00669-f004:**
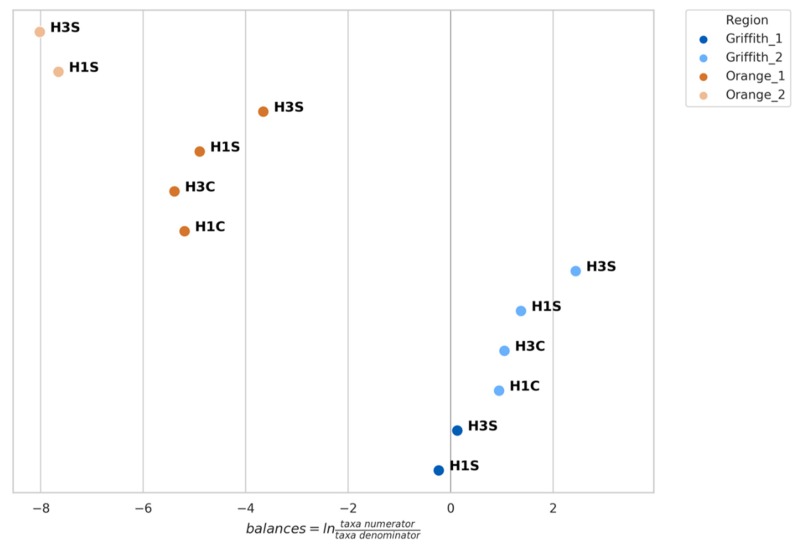
Balances calculated for each sample. Values represent balances median value of replicates.

**Figure 5 microorganisms-07-00669-f005:**
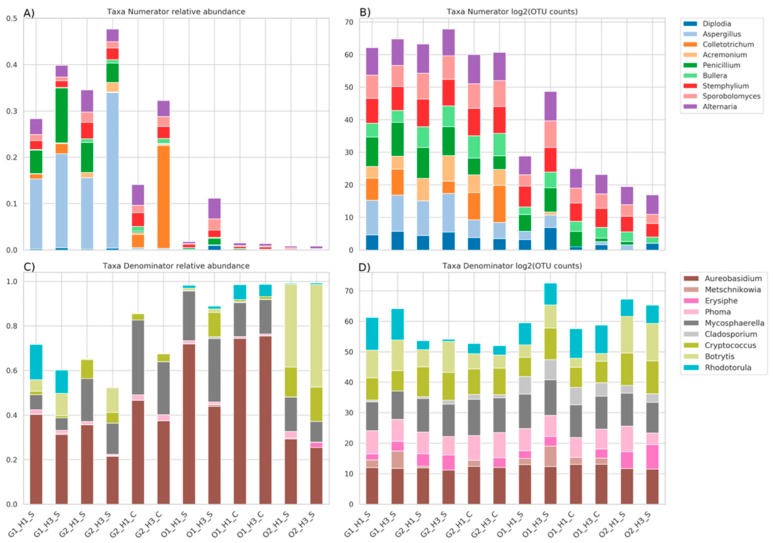
Plots A–C represent relative abundances of the OTU counts for the genera of the rarefied OTU table. Plots B–D represent log2 transformation of the OTU counts for the genera of the rarefied OTU table. Genera have been split into the groups Numerator (**A** and **B**) and Denominator (**C** and **D**) as defined by GNEISS.

**Table 1 microorganisms-07-00669-t001:** Results from Shannon index two-way ANOVA on region O1.

Factor	DF	*R* ^2^	F	Pr (>F)
Variety	1.0	0.25	5.218	0.048
Harvest	1.0	0.29	5.522	0.043

**Table 2 microorganisms-07-00669-t002:** Results from Shannon index three-way ANOVA on all the regions apart region O1.

Factor	DF	*R* ^2^	F	Pr (>F)
Region	2.0	0.53	33.575	1.744e-8
Variety	1.0	0.20	25.710	1.754e-5
Harvest	1.0	0.02	2.807	0.103

**Table 3 microorganisms-07-00669-t003:** Results of the MANOVA analysis performed with ADONIS on Bray–Curtis distance metric.

Factor	DF	R^2^	F	Pr (>F)
Region	3.0	0.532	20.694	0.001
Variety	1.0	0.079	9.267	0.001
Harvest	1.0	0.027	3.229	0.029

**Table 4 microorganisms-07-00669-t004:** OTU counts of rarefied OTU table collapsed at genus level. Values represent median and minimum-maximum range of OTU counts of sample replicates.

Taxonomy	G1_H1_S	G1_H3_S	G2_H1_S	G2_H3_S	G2_H1_C	G2_H3_C	O1_H1_S	O1_H3_S	O1_H1_C	O1_H3_C	O2_H1_S	O2_H3_S
*Diplodia*	25	52	21	45	14	11	9	119	2	3	1	4
	1–46	0–128	12 –33	26–80	6–18	10–11	3–12	66–163	1–6	2–3	0–1	2–5
*Cladosporium*	2	1	2	2	3	2	54	98	53	21	6	7
	1–4	0–3	0–4	1–7	3–6	1–7	21–54	91–135	41–58	19–25	6–7	7–7
*Mycosphaerella*	709	630	2046	1554	3876	2754	2536	3406	1750	1780	1784	1068
	578–1041	467–796	1568–3104	852–2780	3511–4283	2105–2764	1845–2783	3073–3512	1695–1840	1519–2026	1782–2349	1011–1500
*Aureobasidium*	4190	3492	3783	2390	5384	4333	8178	5257	8545	8712	3393	2939
	3373–5254	2849–4175	1996–4907	1502–5085	5237–5559	4122–4625	7963–9262	3293–5927	8513–8696	8331–9090	3289–4590	2868–3802
*Alternaria*	354	284	508	304	517	399	53	536	64	61	47	62
	286–440	206–366	268–804	227–580	440–531	287–428	53–55	415–833	53–64	47–79	44–80	42–72
*Stemphylium*	196	162	376	282	346	300	86	188	49	58	27	18
	107–263	126–232	226–479	207–409	303–404	211–337	86–139	176–214	42–65	35–61	23–48	13–32
*Phoma*	198	147	138	67	279	304	150	118	88	92	341	14
	183–327	118–180	60–319	41–150	266–331	226–316	142–186	89–144	78–92	76–116	70–349	14–23
*Aspergillus*	1567	2263	1634	3734	44	32	6	14	1	2	3	0
	465–2626	1826–3485	234–5548	811–6248	38–67	6–36	5–93	6–35	0–6	1–12	2–4	0–1
*Penicillium*	520	1320	685	460	34	17	35	164	25	2	2	1
	360–943	324–2727	304–1414	268–933	21–54	5–19	16–129	106–307	10–406	1–44	0–4	0–4
*Erysiphe*	4	9	17	30	0	9	6	9	1	8	44	260
	2–8	3–15	7–22	14–83	0–1	5–9	5–12	2–13	0–12	2–9	41–89	251–310
*Botrytis*	540	1121	50	1235	32	21	18	202	8	6	4302	5314
	450–3317	318–1631	10–659	130–1723	22–54	9–25	12–21	132–508	3–9	2–7	2315–4794	4383–5398
*Metschnikowia*	6	51	2	0	4	1	4	103	5	4	0	0
	0–58	0–148	0–5	0–2	2–5	1–4	1–5	16–151	2–6	3–17	0–2	0–1
*Acremonium*	12	16	122	232	44	32	0	2	0	0	1	1
	1–35	1–32	47–662	63–316	43–67	16–33	0–0	1–4	0–2	0–0	0–4	1–1
*Colletotrichum*	112	250	1	14	331	2567	0	0	1	1	0	0
	7–575	42–509	0–2	5–17	187–428	2554–3623	0–1	0–0	0–2	0–1	0–0	0–0
*Rhodotorula*	1638	1153	8	2	10	8	138	131	763	622	47	60
	30–4445	81–2663	0–20	1–10	8–12	6–12	60–252	104–179	512–780	562–716	31–62	45–77
*Sporobolomyces*	139	90	236	148	184	249	11	288	24	22	12	7
	84–264	35–175	89–361	108–311	182–228	127–287	3–14	159–434	18–32	12–41	6–31	5–12
*Bullera*	19	13	84	86	118	125	5	29	9	10	8	4
	12–26	10–17	32–132	51–115	115–128	74–160	0–15	17–35	5–15	4–25	7–9	5
*Cryptococcus*	159	104	902	538	327	410	78	1303	97	136	1559	1770
	81–256	58–170	27–1563	326–1425	277–353	234–468	34–88	691–1739	87–101	110–199	1195–1926	1441–1872

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
