# Peer review of "Geographical Origin Has a Greater Impact on Grape Berry Fungal Community than Grape Variety and Maturation State"

_microorganisms, 2019, doi:10.3390/microorganisms7120669_

Round 1

Reviewer 1 Report

The manuscript “Geographical origin has a greater impact on grape berry fungal community than grape variety and maturation state” represents an interesting contribution to the field of grape and wine microbiomes, focusing on berries.

Abstract is rather confusing. I suggest a complete rewritting.
Introduction
Line 64. Please cite GNEISS authors here or in methods.

Mat and methods
Line 76. Please give coordinates for Orange.
Line 95. Table 1 does not have such information
The use of GNEISS tool is quite interesting but I think that with a replicated experimental design it would be better to apply some differential abundance statistical tool like DESeq2, edgeR…
Results
Line 184-187. Why to include a failure in methods and results? I woud remove bacteria from all manuscript.
Please add some measure of variation to Figure 1
Please use italics for genera names. Please add some measure of variation to table 4
Figure 4 legend says “… the rarified OTU table”, whereas in line 162 says “the unrarefied OTU table” was used as input for GNEISS. Please clarify.
Figure 5 is hard to follow. Anyway, taxa present in numerator plot is absent in denominator plot, so I think that it would be better plot in bars the taxa colored by site, grape and time.

Author Response

The manuscript “Geographical origin has a greater impact on grape berry fungal community than grape variety and maturation state” represents an interesting contribution to the field of grape and wine microbiomes, focusing on berries.

Abstract is rather confusing. I suggest a complete rewritting.

-We will be grateful if the reviewer specifies which part of the abstract is confusing as the rest of the reviewers seemed not to have any problem with it. Nevertheless, we have modified and shortened the abstract and we find it is much clearer now.

Introduction
Line 64. Please cite GNEISS authors here or in methods.

-Thank you for noticing. The reference for GNEISS has been now included. (Morton, J. T., Sanders, J., Quinn, R. A., McDonald, D., Gonzalez, A., Vázquez-Baeza, Y., Navas-Molina, J.A., Song, S. J., Metcalf, J. L., Hyde, E. R., et al. (2017). Balance trees reveal microbial niche differentiation. MSystems, 2(1):e00162–16.)

Mat and methods
Line 76. Please give coordinates for Orange.

-The coordinates of the four vineyards used in the present study have been included now in Supplementary Table 1.

Line 95. Table 1 does not have such information

-Thank you for noticing, it was a mistake and Table 1 should not have been referred there, so we have eliminated it from line 95.

The use of GNEISS tool is quite interesting but I think that with a replicated experimental design it would be better to apply some differential abundance statistical tool like DESeq2, edgeR…

-Differential abundance statistics are known of being problematic for microbial data. DESeq2 and edgeR have been developed for RNASeq gene counts data and the statistical assumptions they make do not necessarily hold for OTU counts since the later are characterized of high sparsity and higher over-dispersion compared to gene counts. The following paper (https://doi.org/10.1186/s40168-017-0237-y) compares various statistical methodologies for differential abundance analysis of microbial data where it shows the weaknesses of DESeq2 and edgeR on this type of data.

For instance, among others the paper mentions the following (we quote):

Furthermore, DESeq and edgeR-trimmed mean by M values (TMM) make the assumptions that most microbes are not differentially abundant, and of those that are, there is an approximately balanced amount of increased/decreased abundance [28]; these assumptions are likely not appropriate for highly diverse microbial environments.

Finally the paper favors the rarefication, which has been used in our study, as a normalization method against other methods including the ones of DESeq and edgeR (we quote):

Rarefying more clearly clusters samples according to biological origin than other normalization techniques do for ordination metrics based on presence or absence. Alternate normalization measures are potentially vulnerable to artifacts due to library size.

 Moreover, before incorporating GNEISS into our analysis, we evaluated the performance of ANCOM (https://www.tandfonline.com/doi/full/10.3402/mehd.v26.27663) which has been developed for differential abundance analysis of microbial data. However, the results were not biologically meaningful since ANCOM was identifying as differential abundant taxa whose counts were fluctuating within a very small range that could be attributed to sequence error. We reasoned the poor performance of ANCOM on the fact that it assumes that only a small number of microbes change across samples. Additionally, we have not reported ANCOM results since this study is not dedicated to benchmarking the performance of ANCOM. In order to do this a more in-depth analysis and different experimental design would be necessary. GNEISS has been developed having in mind these shortcomings of ANCOM and the overall statistical differential  analysis of microbial data by utilizing the concept of balances, that has been used as exploratory analysis in geology since 2005, and without making statistical assumptions. Therefore, the aim of GNEISS is to show informative differences between samples using the balances.  Nevertheless, GNEISS does not provide explicit information regarding differential abundance of taxa.  Therefore the information given from the  balances is not intended to be conclusive, it rather aims to direct the researcher towards a more controlled future experiment in order to study specific microbial changes.

We have incorporated the information of the aforementioned paper in the manuscript. We have added the following lines:

Moreover, the nature of compositional data is known to hinder proper differential abundance analysis since various normalization methods and statistical assumptions could potentially not be appropriate for this type of data.

citation added:  Weiss, S., Xu, Z. Z., Peddada, S., Amir, A., Bittinger, K., Gonzalez, A., Lozupone, C., Zan-eveld, J. R., Vázquez-Baeza, Y., Birmingham, A., et al. (2017). Normalization and microbial differential abundance strategies depend upon data characteristics. Microbiome, 5(1):27.

We altered the information about GNEISS given in the introduction.

GNEIIS introduces the concept of balances which refer to the log-ratio between specific microbial subsets of the community, eliminating the need of using relative abundances and statistical assumptions.

We have introduced a Limitations section where we mention the following:

Finally, GNEISS does not provide an explicit information regarding differential abundance of taxa. As the results demonstrate, it rather servers as an useful exploratory analysis tool aiming at producing comparative insights between the samples utilizing the concept of balances. This way it sets the basis for a subsequent controlled experimental design that focuses to analysing specific microbial dynamics.

Results
Line 184-187. Why to include a failure in methods and results? I woud remove bacteria from all manuscript.

- We understand that the result of the 16S analysis might seem as a failure. It is not a failure but an expected outcome in fact. The massive sequencing of plant material with generic primers for bacteria does not avoid the amplification of DNA of plant organelles like chloroplast or mitochondria. In our case, we used grape berries to extract the DNA and thus, a large percentage of the sequences were classified as Cyanobacteria (ranging from 30000 to 108000 sequences per sample) or Rickettsiales (7400-32000). Those are the closest relatives to the mentioned plant cell organelles. After filtering these host-contaminant sequences, the samples ranged from 99 to 10863 sequences belonging to bacteria (366 on average). That number of sequences is too low to perform a robust analysis of bacterial communities even if the most abundant genus identified after the filtering process (Sphingomonas) was one of the most frequently reported in previous studies about grape berry microbiome. This result indicates that the sequencing was correct but the DNA material from the plant was more abundant and was preferentially sequenced.

Please add some measure of variation to Figure 1

As suggested, we have added bars corresponding to standard deviation to Figure 1. Now the legend of Figure 1 has been modified as follow:

“Samples alpha diversity based on Shannon index. Each value corresponds to the average of samples replicates and bars correspond to standard deviation. Samples abbreviation includes information of the region, Griffith (G1 and G2) or Orange (O1 and O2), the harvest time points (H1 or H3) and the grape varietals, Shiraz (S) or Cabernet (C).”

Please use italics for genera names.

-The italics have been now used for all the genera mentioned along the article. Thank you for noticing.

Please add some measure of variation to table 4

As suggested, we have included the minimum-maximum range of OTU counts for each sample replicate. Now, the legend of table 4 has been modified as follows:

“OTU counts of rarefied OTU table collapsed at genus level. Values represent median and minimum-maximum range of OTU counts of sample replicates.”

Figure 4 legend says “… the rarified OTU table”, whereas in line 162 says “the unrarefied OTU table” was used as input for GNEISS. Please clarify.

-We assume that the reviewer is referring to the legend of Figure 3 since the legend of Figure 4 does not contain such statement. In Figure 3 the only information related to GNEISS are the taxa that are included in the groups numerator and denominator.  GNEISS does not perform any differential analysis, and for that reason after performing GNEISS we wanted to see how the OTU counts could have influenced the balances and which taxa are not statistically significant. In order to do so the difference of sequence depth between the samples should be corrected. And that is why we used the rarefied OTU table.

In lines 271-273 we mention the following:

However, based on the Figure 3 we may assume that the genera that could greatly influence the resulted log-ratio are genera with quite high fold-change between the minimum and maximum observed OTU count.

Although the OTU counts between the rarefied and unrarefied OTU table are different since rarefication leads to power loss, however the relationship between the OTU counts is maintained.

Figure 5 is hard to follow. Anyway, taxa present in numerator plot is absent in denominator plot, so I think that it would be better plot in bars the taxa colored by site, grape and time.

-Unfortunately, we fail to understand the suggestion of the reviewer on how the information of Figure 5 should be depicted.

Regarding the taxa of the numerator and the denominator we have specified on the legend of Figure 5 that correspond to the groups numerator and denominator as defined by GNEISS.

In lines 193-197 we mention the following:

“Each group contains genera that are highly correlated based on their co-occurrence and therefore the two groups are anti-correlated. Subsequently, GNEISS applied isometric log-ratio transformation which calculates, for each sample, the log-ratio between these two groups.  That means one group represents the numerator of the ratio and the other group the denominator, whereas the log-ratio is referred as balance. ”

Since the taxa of the numerator and the taxa of denominator are anti-correlated, that means that taxa from the numerator group cannot be in the denominator group.

In Figure 5 we have plotted the OTU counts as relative abundance and as log2 values.

Relative abundance is a typical representation of OTU counts applied in other studies. However relative abundance may lead to misinterpretations as we mention in lines 58-60:

One of the major drawbacks of the studies analyzing microbiome diversity by means of HTS techniques is the derivation of statistical inferences after converting the OTU (operational taxonomic unit) counts of the identified genera to relative abundance. The transformation of the OTU counts to compositional data on one hand adds the constraint of the abundances having to sum to a constant (i.e. 1), and on the other hand may lead to misinterpretations when multivariate statistics are applied [24].

In lines 84-85 we mention the following:

GNEIIS introduces the concept of balances which refer to the log-ratio between specific microbial subsets of the community, eliminating the need of using relative abundances.

In lines 274-275 we mention the following:

Figure 5 shows the abundances of the identified genera based on their observed OTU counts. The OTU counts have been converted to relative abundance in order to be compared to their log2 values.

Therefore in Figure 5 relative abundance is compared to log2 values. In lines 373-379  we mention the following:

From the comparison between relative abundance and log2 OTU counts it becomes  apparent that the constraint applied when the OTU counts are converted to compositional data could lead to misinterpretations. For instance, the fluctuation of the relative abundance of Aureobasidium across all samples seems significant. However, the log2 transformation of the OTU counts of  Aureobasidium reveals a relatively stable abundance across all samples corroborating its identification as non-significant genus by the current analysis.

In lines 282-284 we mention the following:

The log2 transformation of the OTU counts in Figure 5 will be used as a mean of explaining the calculated balances depicted in Figure 4, taking into consideration the aforementioned genera of Figure 3 that could greatly influence the resulted balance.

Finally, the annotation of the samples on the x-axis of Figure 5 has been remained the same as the annotation of the samples in Figure 1 in order to facilitate comparisons and avoid confusions.

Reviewer 2 Report

In this work, the authors perform an experimental investigation by means of barcoded massive sequencing on Cabernet Sauvignon and Shiraz grapes varieties sampled at two ripening times in four vineyards situated in two different Australian wine regions. The aim is to establish the effect of the grape ripening state, the vineyard region and grape variety on the grape mycobiome.

The main findings suggest that the vineyard region mainly effects both fungal composition and diversity; on the contrary, the grape variety and the ripening state have a minor contribution on that. However, since the ripening state depends on climate conditions, this will affect the corresponding fungal communities. Hence, a special care should be taken when analyzing the fungal composition.

The paper is sound, interesting and well written. The results, reinforced by a proper statistical analysis, put new insights into the relation between the grape characteristics (ripening state, vineyard region and variety) and the resultant mycobiome.

The authors, after a care reading, should revise the paper for some faults and few other details. For example:

they inserted wrong geographical coordinates for the Griffith region (page two, they should read 34°17'00''S 146°02′00''E) but no coordinates for the Orange regions are given. If appropriate, geographical coordinates should be provided for all the considered vineyards. Section 3.4 is missing and section 3.5 has a bigger font. A paragraph describing the error analysis and the precision of the obtained results is missing

For these reasons, in my opinion the manuscript deserves publication on Microorganisms after addressing the points raised above.

Author Response

In this work, the authors perform an experimental investigation by means of barcoded massive sequencing on Cabernet Sauvignon and Shiraz grapes varieties sampled at two ripening times in four vineyards situated in two different Australian wine regions. The aim is to establish the effect of the grape ripening state, the vineyard region and grape variety on the grape mycobiome.

The main findings suggest that the vineyard region mainly effects both fungal composition and diversity; on the contrary, the grape variety and the ripening state have a minor contribution on that. However, since the ripening state depends on climate conditions, this will affect the corresponding fungal communities. Hence, a special care should be taken when analyzing the fungal composition.

The paper is sound, interesting and well written. The results, reinforced by a proper statistical analysis, put new insights into the relation between the grape characteristics (ripening state, vineyard region and variety) and the resultant mycobiome.

The authors, after a care reading, should revise the paper for some faults and few other details.

-The paper has been now carefully revised and some faults included those noticed by the reviewer have been now corrected. Thank you a lot for noticing.

For example:

they inserted wrong geographical coordinates for the Griffith region (page two, they should read 34°17'00''S 146°02′00''E) but no coordinates for the Orange regions are given. If appropriate, geographical coordinates should be provided for all the considered vineyards.

-The coordinates for the Griffith region has been modified as suggested by the reviewer and it has been included now in a supplementary table (Supp. Table 1) together with the coordinates and elevation (m a.s.l) of the rest of vineyards used during the study.

Section 3.4 is missing and section 3.5 has a bigger font.

-Thanks for noticing, we have corrected it accordingly.

A paragraph describing the error analysis and the precision of the obtained results is missing

-Indeed, an error analysis would be quite useful in order to evaluate the obtained results. However, such analysis would require the comparison between observed and expected values. In a metataxonomic exploratory analysis of samples such as those in our study the taxonomy and the microbial abundance is unknown. Moreover, various laboratory and bioinformatic procedures are involved each one associated with many factors that add variability among the samples. Therefore, pinpointing the amount of variability produced among the samples is rendered difficult. Nevertheless, we have incorporated a limitations section in our manuscript that addresses these issues (current lines 381-394).

For these reasons, in my opinion the manuscript deserves publication on Microorganisms after addressing the points raised above.

Reviewer 3 Report

I have few question and minor spelling-related suggestions. Please find them bellow.

Line 53: Please provide full-form of UFC

Line 58: Please correct the spelling of major

Why 16S sequences were low in number? Do authors have any explanation for this?

Aureobasidium and Mycosphaerella were the most abundant genera, ubiquitous across the samples without any significant role to play. Is it possible that these two fungi live as endophytes and share some kind of symbiotic relationship with grapes? 

Author Response

I have few question and minor spelling-related suggestions. Please find them bellow.

Line 53: Please provide full-form of UFC

-The full form of CFU (Colony Forming Units) has now been provided and the UFC have been changed by CFU at the same line.

Line 58: Please correct the spelling of major

-The spelling mistake has been corrected as suggested.

Why 16S sequences were low in number? Do authors have any explanation for this?

-The massive sequencing of plant material with generic primers for bacteria does not avoid the amplification of DNA of plant organelles like chloroplast or mitochondria. In our case, we used grape berries to extract the DNA and thus, a large percentage of the sequences were classified as Cyanobacteria (ranging from 30000 to 108000 sequences per sample) or Rickettsiales (7400-32000). Those are the closest relatives to the mentioned plant cell organelles. After filtering these host-contaminant sequences, the samples ranged from 99 to 10863 sequences belonging to bacteria (366 on average). That number of sequences is too low to perform a robust analysis of bacterial communities even if the most abundant genus identified after the filtering process (Sphingomonas) was one of the most frequently reported in previous studies about grape berry microbiome. This result indicates that the sequencing was correct but the DNA material from the plant was more abundant and was preferentially sequenced.

Aureobasidium and Mycosphaerella were the most abundant genera, ubiquitous across the samples without any significant role to play. Is it possible that these two fungi live as endophytes and share some kind of symbiotic relationship with grapes? 

-As commented in the discussion of the article, most of Mycosphaerella species has been reported as plant pathogens. However, it would be a possible that some species of the genera were endophytes. In fact, Dissanayake et al. (2018) identified both genera, Aureobasidium and Mycosphaerella, within the endophytic community in stems grapevine. We have now included this comment in the text to acknowledge the possibility. Thank you for the suggestion.

Round 2

Reviewer 1 Report

After corrections I think that the manuscript can be accepted